# Heterogeneity of Response to Iron-Based Metallodrugs in Glioblastoma Is Associated with Differences in Chemical Structures and Driven by FAS Expression Dynamics and Transcriptomic Subtypes

**DOI:** 10.3390/ijms221910404

**Published:** 2021-09-27

**Authors:** Anne Vessières, Emie Quissac, Nolwenn Lemaire, Agusti Alentorn, Patrycja Domeracka, Pascal Pigeon, Marc Sanson, Ahmed Idbaih, Maïté Verreault

**Affiliations:** 1Institut Parisien de Chimie Moléculaire, Sorbonne Université, CNRS, UMR CNRS 8232, 4 Place Jussieu, F-75005 Paris, France; pascal.pigeon@chimieparistech.psl.eu; 2Institut du Cerveau-Paris Brain Institute-ICM, Inserm, Sorbonne Université, CNRS, APHP, Hôpital de la Pitié Salpêtrière, F-75013 Paris, France; emie.quissac@icm-institute.org (E.Q.); nolwenn.lemaire@icm-institute.org (N.L.); pat.domeracka@gmail.com (P.D.); 3Institut du Cerveau-Paris Brain Institute-ICM, Inserm, Sorbonne Université, CNRS, APHP, Hôpital de la Pitié Salpêtrière, DMU Neurosciences, Service de Neurologie 2-Mazarin, F-75013 Paris, France; agusti.alentorn@aphp.fr (A.A.); marc.sanson@aphp.fr (M.S.); ahmed.idbaih@aphp.fr (A.I.); 4Chimie ParisTech-PSL, 11 Rue P. et M. Curie, F-75005 Paris, France

**Keywords:** ferrocene, death receptor signaling pathway, biomarkers, targeted therapy, personalized medicine, bioorganometallic chemistry

## Abstract

Glioblastoma (GBM) is the most frequent and deadliest primary brain cancer in adults, justifying the search for new treatments. Some members of the iron-based ferrocifen family have demonstrated a high cytotoxic effect on various cancer cell lines via innovative mechanisms of action. Here, we evaluated the antiproliferative activity by wst-1 assay of six ferrocifens in 15 molecularly diverse GBM patient-derived cell lines (PDCLs). In five out of six compounds, the half maximal inhibitory concentration (IC_50_) values varied significantly (10 nM < IC_50_ < 29.8 µM) while the remaining one (the tamoxifen-like complex) was highly cytotoxic against all PDCLs (mean IC_50_ = 1.28 µM). The pattern of response was comparable for the four ferrocifens bearing at least one phenol group and differed widely from those of the tamoxifen-like complex and the complex with no phenol group. An RNA sequencing differential analysis showed that response to the diphenol ferrocifen relied on the activation of the Death Receptor signaling pathway and the modulation of FAS expression. Response to this complex was greater in PDCLs from the Mesenchymal or Proneural transcriptomic subtypes compared to the ones from the Classical subtype. These results provide new information on the mechanisms of action of ferrocifens and highlight a broader diversity of behavior than previously suspected among members of this family. They also support the case for a molecular-based personalized approach to future use of ferrocifens in the treatment of GBM.

## 1. Introduction

In adults, glioblastoma (GBM) is the most frequent primary brain cancer, accounting for nearly 50% of gliomas [1,2,3]. In the vast majority of cases, the outcome of GBM patients remains dismal with a median overall survival (OS) ranging between 12 and 24 months despite various intensive treatments including surgical resection, cytotoxic or targeted chemotherapy and radiation therapy [4,5,6]. New therapeutic approaches are clearly needed to improve the prognosis of GBM patients.

Inter-tumoral heterogeneity in GBM, characterized through genomic and transcriptomic profiling, has been suggested as being partly responsible for heterogeneous patient prognosis [7]. Over recent decades, identification of recurrent molecular abnormalities and disrupted intracellular signaling pathways in GBM has improved our understanding of oncogenic drivers in these tumors and their biological and clinical impact [8]. For example, the *TP53* gene, which controls cell proliferation, survival, and genome integrity, is mutated in 28% of GBM cases. Such alterations have been associated with a decrease in chemosensitivity and worse prognosis in patients [9,10]. Some studies have further integrated genomic and transcriptomic information and have helped to establish a comprehensive view of the GBM molecular landscape and its impact on clinical behavior [11]. The most recent version of molecular subtyping divides GBM into three different transcriptomic subtypes: Proneural, Mesenchymal, and Classical [12], with the Mesenchymal subtype being associated with poorer prognosis.

Ferrocifens are a family of iron-based organometallic complexes having in common a ferrocenyl–double bond–phenol (or phenyl) motif (Figure 1) [13,14]. Complex **P15**, the first complex studied in this series, is the ferrocenyl derivative of hydroxytamoxifen, the active metabolite of tamoxifen, which is the standard antiestrogen used to treat hormone dependent breast cancer [14]. **P15** and its diphenol equivalent **P5** both showed a strong and comparable antiproliferative effect on the triple negative cancer cell line MDA-MB-231 (IC_50_ = 0.5 and 0.6 µM, respectively) [15]. Their in vivo efficacy has also been established, for **P15** on tumor xenografts of MDA-MB-231 cells in mice [16] and for **P5**, on orthotopic or ectopic tumors of 9L GBM in rats [17,18]. The principal mechanism of action of ferrocifens to account for this strong cytotoxic effect has been linked to the reversible oxidation of Fe(II) to Fe(III) (Fenton and Haber–Weiss reactions) leading to the production of quinone methides, reactive molecules that can induce cell death [19,20,21,22]. Ferrocifens also possess the unusual intracellular behavior of generating reactive oxygen species (ROS) [13,23], which are known to cause cell damage [24]. However, the mechanisms of action of these complexes are far from fully understood, particularly in terms of the signaling pathways involved.

Here, we evaluated the efficacy of a series of six representative members of the ferrocifen family (**P5**, **P15**, **P41**, **P53**, **P85**, **P722**; Figure 1) selected from the 300 ferrocifens synthesized previously [13,21] in a panel of 15 GBM patient-derived cell lines (PDCLs) that were selected to represent the heterogeneity of GBMs encountered in the clinic (Table 1). **P15** allowed us to assess the role of the amino chain on compound efficacy. The series **P5**, **P85**, **P41** (complexes with 2, 1 or no phenol substituent) gave access to the role of phenol groups, while **P53** and **P722** were selected to evaluate the effects of modifications of the lateral chain.

The aim of the work presented here is twofold. First, we evaluated the antiproliferative effects of these molecules on the PDCL panel, with the hope of providing evidence of the heterogeneity of response to the compounds in different cell lines. Then, through an RNA sequencing differential analysis performed on **P5**, which was selected as being representative of a group of four compounds sharing similar behavior, we sought to identify mechanisms and biomarkers of response to this class of compounds. In the field of metallo-drugs, Kim et al. have identified molecular markers for oxaliplatin, one of the three platinum-based inorganic complexes widely used in cancer chemotherapy, in 14 PDCCEs (patient-derived colorectal cancer explants) [25]. However, to the best of our knowledge, the work presented here is the first study of the effects of a selection of organometallic complexes on a panel of PDCLs, in this case from GBM patients.

## 2. Results

### 2.1. Ferrocifens Induce Diverse Impacts on PDCL Viability According to the Genetic Context

The ferrocifens used in this study (**P5**, **P15**, **P41**, **P53**, **P85**, **P722**; Figure 1) were synthesized as described previously (see Materials and Methods). For this study, a panel of 15 GBM PDCLs was selected (Table 1). These GBM PDCLs are part of the GlioTEx cell line bank (ICM, Paris Brain Institute), and they are cultured under neurosphere conditions in the absence of serum in order to preserve the phenotype and genotype of parental tumors [26,27]. The panel of PDCLs was assembled to represent a heterogeneous cohort at the molecular level (Table 1). Approximately half (8/15) of the selected PDCLs carry a mutant form of TP53. PDCLs were also classified according to transcriptomic subtypes [11]. Twenty-six percent (4/15) of the PDCLs are classified as Proneural, thirty-three percent (5/15) are from the Mesenchymal subtype, and forty percent (6/15) are from the Classical subtype.

#### 2.1.1. IC_50_ Values of Ferrocifens on the Panel of PDCLs

Figure 2A shows the IC_50_ values for each compound and PDCL compared to the mean IC_50_s obtained for all PDCLs, allowing identification of sensitive (to the left of the axis) and resistant PDCLs (to the right of the axis) and evidencing a large difference in the behavior of the complexes. The 90 IC_50_s are listed in Appendix A. They covered a wide range of values (10.0 nM < IC_50_ < 29.7 µM), with IC_50_ mean values of the complexes as follows: **P15**< **P53** < **P41** < **P5** = **P85** < **P722**. Compound **P15**, the complex with the tamoxifen-like amino side chain, was found to be active against all PDCLs. It had the smallest mean IC_50_ (1.28 µM) and the smallest IC_50_ max/IC_50_ min ratio (7.1) of the six compounds (Figure 2B,C), indicating a strong and homogeneous antiproliferative effect on survival/proliferation of all PDCLs. In contrast, the diphenol complex **P5** showed a 50-fold greater IC_50_ max/IC_50_ min ratio (350), corresponding to a high heterogeneity of action against the panel of PDCLs. The IC_50_ min obtained with **P5** was ~seven times smaller than that of **P15** (70 nM versus 0.52 µM). The diphenol and monophenol complexes, **P5** and **P85**, behaved similarly in our panel (Figure 2B,C and Appendix A), although **P5** IC_50_ min was 14 times lower than that of **P85**, which resulted in a much larger IC_50_ max/IC_50_ min ratio for **P5**. Complex **P41**, with no phenol function, behaved differently from **P5** (Figure 2B,C and Appendix A) with a mean IC_50_ two times lower (4.54 versus 8.91 µM) and IC_50_ values more homogeneous. This result, indicating a global toxicity higher for **P41** than for **P5** against these PDCLs, was unexpected based on our previous data on breast cancer cells (MDA-MB-231) where **P41** was significantly less cytotoxic than **P5** (7.5 and 0.64 µM, respectively) [15].

Regarding **P53**, the complex with a hydroxypropyl side chain, while the mean IC_50_ was two times smaller than that of **P5**, the IC_50_ min was four times higher. Finally, **P722**, the complex with an imido propyl side chain, showed the highest and the lowest IC_50_ values of the series (10.00 nM and 29.77 µM, respectively), resulting in the largest IC_50_ max/IC_50_ min (~3000). Interestingly, a positive correlation was found between the IC_50_ values of **P5 vs. P722** or **P53** (Appendix A), suggesting similarities in the mechanisms of action for these compounds.

#### 2.1.2. IC_50_ Values According to the PDCLs

Figure 3A,B highlight the heterogeneity of response of PDCLs to ferrocifens. Three PDCLs out of 15 showed large IC_50_ max/IC_50_ min ratios (higher than 200; PDCL 2, 5, and 6). In contrast, 3/15 PDCLs showed small IC_50_ max/IC_50_ min ratios (lower than 10; PDCL 15, 9, and 10). No correlation could be seen with the presence of *TP53* mutations. Next, the response of PDCLs to all six compounds was analyzed compared to the mean IC_50_ obtained for each compound (Figure 3C). In five PDCLs (PDCL 2, 1, 5, 15, and 11), the IC_50_ values were lower than the mean IC_50_ for all compounds. These PDCLs were therefore considered sensitive to all compounds. In four PDCLs (PDCLs 8, 12, 14, and 4), the IC_50_s were higher than the mean IC_50_ for all compounds, and these PDCLs were considered resistant. Finally, a group of six PDCLs (PDCLs 7, 3, 6, 9, 10, and 13) showed a heterogeneous response to the different compounds. Again, no correlation was found with the presence of a mutant form of *TP53* within these three subgroups, suggesting that *TP53*-dependant apoptosis is not crucial in the mechanism of response to these compounds.

### 2.2. Death Receptor Signaling and FAS Expression Dynamics Predict the Response to P5

We thus undertook a comparison of the RNA expression profiles of ferrocifen-treated and untreated PDCLs to identify key molecular pathways predictive of response. **P5** is the ferrocifen most studied so far [13,21]. In addition, we showed here that its pattern of response was similar to that of three other ferrocifens in our PDCLs (**P85**, **P53**, **P722**; Appendix A). Therefore, **P5** was selected to perform a mechanistic study in one sensitive (PDCL 1^Sens^) and one resistant (PDCL 12^Res^) PDCL.

PDCL 1^Sens^ and PDCL 12^Res^ were exposed to **P5** at their respective IC_30–50s_ (0.38 and 5 µM) for 24 h. In both PDCLs, the top pathways activated were linked to the control of the cell cycle. Importantly, in the sensitive PDCL 1^Sens^, the Death Receptor (DR) signaling pathway was activated, which ultimately led to apoptosis (Figure 4A,B). This pathway was not activated in the resistant PDCL 12^Res^. To validate this observation in additional models, the expression of two members of the Death Receptor pathway (TNFR2 and FAS) was evaluated by RT-qPCR in three resistant (PDCLs 12, 13, and 4) and three sensitive (PDCLs 1, 5, and 6) PDCLs in response to **P5**. Interestingly, while the expression of FAS was significantly lower in sensitive PDCLs under basal conditions compared to resistant PDCLs, it was found to be consistently elevated in all sensitive PDCLs when they were exposed to **P5**, in contrast to resistant PDCLs (Figure 4C). Changes in the expression of TNFR2 were not significantly different between sensitive and resistant PDCLs (Appendix A). Increased expression of FAS as well as other members of the Death Receptor signaling pathway (DR6, TNFR2, and Caspase 8) was confirmed at the protein level in the sensitive PDCL 1^Sens^ in response to **P5** (Figure 4D). Notably, the expression of proteins of the Bcl-2 family (Bax, Bad, Bcl-2, Bid, Bcl-w) or *TP53* signaling pathway (p53, p21) was found to be lower or unchanged, suggesting that these canonical apoptosis pathways are not crucial in **P5**-induced apoptosis.

### 2.3. Transcriptomic Subtypes Predict the Response to P5 and P53

Finally, we performed correlations between the transcriptomic subtyping of PDCLs (Table 1) and their response status to ferrocifens in an attempt to identify the biomarkers of response. Importantly, correlations between the PDCL IC_50_ values and their score for the Classical subtype showed that both **P5** and **P53** were less active in PDCLs with the highest Classical subtype score (Figure 5A). Indeed, we found that the majority of PDCLs resistant to **P5** (Δ from the mean IC_50_ ˃ 2 µM) belonged to the Classical subtype, while the sensitive PDCLs (Δ ˂ 2 µM) were classified either in the Proneural or in the Mesenchymal subtypes (Figure 5B,C). Similarly, PDCLs resistant to **P53** (Δ ˃ 1 µM) mainly belonged to the Classical subtype, while the sensitive PDCLs (Δ ˂ 1 µM) were classified either in the Proneural or in the Mesenchymal subtypes (Figure 5C). A non-significant trend was observed for **P85** (Figure 5A). No correlation was found between any members of the transcriptomic subtype and IC_50_s for **P15**, **P722,** and **P41**.

## 3. Discussion

The IC_50_ values obtained for the six ferrocifens on the GBM PDCL panel, as well as the correlation of the IC_50_ values for **P5** with the other complexes (Figure 2 and Appendix A), allowed them to be classified into three different groups, G1 (**P5**, **P85**, **P53**, **P722**), G2 (**P15**), and G3 (**P41**).

The tamoxifen-like complex **P15** stands out from the others for its high cytotoxicity against all PDCLs (Figure 2) and the lack of correlation between its IC_50_ values and those of **P5** (Appendix A). The great difference in behavior between **P5** and **P15**, linked to the presence of an aminoalkyl chain in **P15**, can be explained by the fact that at physiological pH, this chain is protonated. This confers on **P15** the status of “lipophilic cation”, an entity known to cause depolarization of the mitochondrial membrane followed by cell death [29]. This is essentially what was observed previously with **P15** in Jurkat cells [30]. An interaction of **P15** with mitochondria seems therefore to be the mechanism of action governing the effect of **P15** in this cell panel. This result also reveals that the mechanisms of action of **P5** and **P15**, originally thought to be identical due to their similar IC_50_ values for MDA-MB-231 cancer cells (0.5 and 0.64 µM for **P15** and **P5**) [15], are in fact quite different from each other. The comparable toxicity first observed for **P5** and **P15** against breast cancer cells has been attributed to the redox effect specific to the ferrocenyl–double bond–phenol chain, which is shared by these molecules and leads to the formation of quinone methides (QM), reactive species that can react with nucleophiles and thus cause cell death [19,20,22]. In agreement with our findings here, however, a number of studies have subsequently shown that the behavior of these compounds can differ. For example, in melanoma cell lines, **P5** and **P15** showed totally different patterns of toxicity, with **P15** being more potent after a short incubation time at high concentration, while the cytotoxic effect of **P5** required a lower concentration over a longer period of time [31]. We subsequently showed that **P15**, but not **P5**, was able to inhibit thioredoxine reductase (TrxR) in Jurkat cells [32]. This result may be explained by the transformation of the QM of **P5** to the corresponding indene, a molecule that is unable to inhibit TrxR [21,33]

The G1 group (**P5**, **P85**, **P53**, **P722**) encompasses all the complexes that share the ferrocenyl–double bond motif, at least one phenol group, and no tamoxifen-like chain. There was a correlation between the IC_50_ values of **P5** and the other members of this group (Appendix A), indicating that these compounds share one or more mechanisms of action associated with their cytotoxicity. There was a better correlation between **P5** and **P53** or **P722**, the two complexes with modified side chains, than with **P85**, the complex with a single phenol. The presence of two phenols therefore seems to play an important role, and this is in agreement with the fact that it has recently been shown that only the trans (and not the *cis*) ferrocenyl–double bond–phenol alignment allows the complex to induce cytotoxicity [33]. This is indeed always present in the diphenol compounds, while it is only present in 50% of the monophenol **P85** due to the *cis*/*trans* isomerization of the complexes [33]. Finally, the modification of the side chain appears to have a significant effect on the heterogeneity of the response generated by the complexes (Figure 2). Indeed, **P722**, the complex bearing an imido propyl chain, was the one with the highest and lowest IC_50_ values of the entire panel, while **P53**, which has a hydroxypropyl chain, showed a lower heterogeneity of response on the PDCL panel. This may be explained by the difference in structure, and thus in reactivity, of their QMs. In fact, the QM of **P722** is stabilized via a lone-pair/pi interaction [34] while **P53** can lead to a QM bearing a THF (tetrahydrofuran) ring. This modifies its reactivity since **P53-QM** can only lead to 1,6 Michael additions, while with **P5**, only 1,8 Michael additions are possible [35]. An in-depth study of the biological properties of **P53** and **P722** will be undertaken shortly.

As for **P41**, the complex bearing a phenyl substituent in place of the phenol, the non-correlation of its IC_50_ values with those of **P5** (Appendix A) is interesting. In fact, **P41**, which does not contain a phenol group, cannot lead to the formation of QM, the entity believed to be the source of its toxicity, as mentioned above. This is the favored explanation for the fact that in MDA-MB-231 cells, **P41** was significantly less cytotoxic than **P5** (IC_50_ = 7.5 and 0.64 µM, respectively) [15]. However, the fact that **P41** showed higher toxicity than **P5** in five PDCLs of this panel cannot be attributed to a QM-related mechanism of action. An explanation for this result is to be found in the general scheme of ferrocifen cytotoxicity, which shows that Reactive Oxygen Species (ROS) are rapidly formed in vitro only a few minutes after the complexes enter MDA-MB-231 cells [13,23]. Indeed, it has been observed that **P41** is the compound that produces the most ROS in these cells [13]. Although it has been shown that the formation of ROS plays a role in the cytotoxicity of the ferrocifens, with the addition of the antioxidant *N*-AcetylCysteine reducing the toxicity of **P5** and **P15** [33,36], we noted that in MDA-MB-231 cells, there was no correlation between ROS production and cytotoxicity [13]. It is, however, widely accepted that ROS are able to trigger programmed cell death in cancer cells [24]. It appears therefore that this cell death mechanism is particularly effective in this sub-group of PDCLs. This result is consistent with studies by Osella et al. showing that the cytotoxicity of ferrocenium salts is associated with the formation of ROS [37]. This is also the case for two organometallic complexes of iridium and osmium, recently studied by Sadler et al. [38,39], as well as for doxorubicin, the production of ROS in that case being associated with the presence of intracellular iron [40].

When looking at the molecular context present in PDCLs sensitive to ferrocifens, it appears that the presence of mutations on TP53, a key regulator of apoptosis, does not block the effect of ferrocifens, suggesting that TP53-dependent apoptosis pathways are not crucial in the response to ferrocifens. In fact, we found here that a low baseline level of FAS expression and an increased FAS expression when PDCLs were exposed to the compound were associated with the response to **P5**. FAS (also called CD95) is a member of the tumor necrosis factor superfamily and is involved in the regulation of cell death and survival (reviewed in [41]). Ligation of Fas ligand (FasL) to its receptor will result in receptor trimerization and the recruitment of molecules such as Fas-associated death domain (FADD) and caspase-8 to form the CD95 death-inducing signaling complex (DISC). Caspase-8 will then be activated by self-cleavage and will, in turn, activate downstream effector caspases such as caspase-3. The CD95 system has been implicated in tumor cell death induced by a number of chemotherapies such as the DNA-damaging agents doxorubicin or cisplatin [41,42] or the DNA synthesis inhibitor 5-FU [43]. Corresponding to what we report here for ferrocifens, its expression was also shown to play a role in determining sensitivity or resistance to some of these chemotherapies [44].

Deficient activation of the CD95 pathway and failure to activate the caspase effectors in PDCLs resistant to ferrocifens could be due to several factors such as an imbalanced expression of apoptosis-modulating proteins [45]. Interestingly, we found a correlation between the resistance of PDCLs to **P5**/**P53** and their membership of the Classical transcriptomic subtype (Figure 5). This subtype is characterized by activation of RTK signaling pathways such as EGFR [11]. Overexpression of EGFR was previously reported to induce resistance to FasL-induced cell death [46]. Conversely, FasL-induced cell death was enhanced by EGFR inhibition using tyrphostine [46]. The authors suggested that this cross-talk could involve the activation of EGFR downstream signaling proteins such as AKT and c-FLIP, which can in turn modulate the activity of caspases. Hence, in our PDCLs from the Classical subtype, it is possible that RTK signaling altered the balance of apoptosis-modulating proteins towards a state of resistance to FAS-induced cell death. Conversely, mechanisms regulating FAS-induced cell death in sensitive PDCLs from the Proneural and Mesenchymal subtypes may be less influenced by RTK signaling and thus more functional. Notably, the Mesenchymal PDCL resistant to **P5** (PDCL 12) expresses the FAS Exo8 Del alternative FAS variant, which is known to block FAS-mediated apoptosis [47]. It is thus possible that in this PDCL, this variant may have contributed to resistance to **P5** in a molecular context that would have been otherwise favorable to FAS-induced cell death.

Importantly, we found that amongst our *TP53* mutant PDCLs, four of eight responded to all ferrocifen compounds (Figure 2C), with 6/8 being sensitive to **P5** and **P53** and having IC_50_s lower than 10 µM. *TP53* mutations have been associated with worse prognosis in patients [9] and resistance to temozolomide in vitro [48]. Moreover, we found that most PDCLs from the Mesenchymal subtype were responders to **P5** and **P53**. The Mesenchymal subtype has also been associated with worse prognosis in patients and resistance to temozolomide in vitro [49]. Interestingly, **P15** was active against all PDCLs regardless of their transcriptomic subtype or *TP53* mutational status and could constitute an effective therapeutic option for all GBM. Thus, ferrocifens provide an attractive alternative to treat resistant GBM.

The value of individualized therapy for GBM has been recognized by the community [50,51,52], although the benefit in terms of improved patient outcomes has been hindered, in part, by cellular resistance mechanisms emerging in response to targeted therapies [53]. The identification of biomarkers of response to temozolomide [54], the chemotherapy used in the standard of care [4], has improved our knowledge of the mechanism of action of this drug, but the lack of alternative treatments for resistant (non MGMT methylated) GBMs has limited the clinical implications of this finding [55]. Our work provides another approach for personalized medicine in the context of metallodrugs. Oxaliplatin is a good example of the importance of individual assessment of tumor response, as only 40–45% of colorectal cancer patients would benefit from oxaliplatin-based therapy [25]. Thus, several studies have been published with the aim of finding molecular markers of sensitivity to optimize oxaliplatin treatment outcome for these patients [25,56].

Here, we report a diversity of behaviors both amongst ferrocifens and between GBM PDCLs linked to differences in chemical structures and genetic contexts, shedding new light on the potential of ferrocifens in the era of personalized medicine.

## 4. Materials and Methods

Further information and requests for resources and reagents should be directed to the Lead Contacts Anne Vessieres (anne.vessieres@sorbonne-universite.fr) and Maïté Verreault (maite.verreault@icm-institute.org). A Material Transfer Agreement may be required.

### 4.1. Synthesis of the Compounds

Ferrocifens were prepared as described in the following publications: **P15** [14], **P5** [57], **P85** [58], **P41** [59], **P53** [60], **P722** [34]. Stock solutions (10^−3^ M) of ferrocifens were prepared in DMSO. The maximal final concentration of DMSO in cell culture was 1%, allowing efficient accumulation of the compounds in cells [30].

### 4.2. Cell Lines

All GBM PDCLs were established by the GlioTEx team (Glioblastoma and Experimental Therapeutics) in the Paris Brain Institute (ICM) laboratory and maintained at 37 °C, 5% CO_2_ under neurosphere growth conditions using DMEM/F12 (Gibco, Life Technologies, Saint-Aubin, France) culture medium supplemented with 1% penicillin/streptomycin, B27 diluted 1:50 (Gibco), EGF (20 ng/mL), and FGF (20 ng/mL) (Preprotech, Neuilly-sur-Seine, France). The identity of all cell lines established at the ICM was confirmed by short tandem repeat (STR) assay according to the manufacturer’s instructions (PowerPlex 16, Promega, Charbonnières-les-Bains, France). PCR products were sent to Genoscreen (Lille, France) to determine STR profiles. The profiles were compared to the parental tumors and validated within three months of their use for the studies presented here.

### 4.3. Cell Viability Assay

All tests described below were performed in at least three independent experiments: 96-well plates were coated with 10 µg/mL laminin (#L2020, Sigma–Aldrich, Saint-Quentin Fallavier, France) at 37 °C for 1 h. Three thousand cells/well were then plated in full culture medium. Compounds were added the next day from 1 mM stock solutions in DMSO. Seventy-two hours later, cell viability was assessed using WST-1 reagent (Roche, Meylan, France) according to the manufacturer’s instructions.

### 4.4. RNA Sequencing and Analysis and TP53 Mutation Status

RNA expression profiles were acquired for each PDCL under basal conditions or, for PDCL 1^Sens^ and PDCL 12^Res^, after a 24-h exposure to their respective **P5** IC_30–50_ (0.38 and 5 µM, respectively) or culture medium. For basal conditions, PDCLs were dissociated on day—3, the culture medium was replaced on day—1, then cells were harvested on day 0. RNA was extracted using a Qiagen (Courtaboeuf, France) RNeasy mini-kit. mRNAseq libraries were prepared using an mRNA stranded library preparation kit and sequenced with an ILLUMINA system (Nextseq 500 or Novaseq 6000). Library preparations were performed following manufacturer’s recommendations and then sequenced to obtain a minimum of 2*30 million reads per sample. The quality of the raw data was evaluated with FastQC. Poor quality sequences were trimmed or removed with Fastp software to retain only good quality paired reads. Star v2.5.3a was used to align the reads against hg19 reference genome using default parameters except for the maximum number of multiple alignments allowed for a read, which was set to 1. Quantification of gene and isoform abundances was done with the rsem 1.2.28 on RefSeq catalogue prior to normalization with the edgeR bioconductor package. Finally, differential analysis was conducted with the generalized linear model (GLM) framework likelihood ratio test from edgeR. Multiple hypothesis adjusted *p*-values were calculated with the Benjamini–Hochberg procedure to control FDR. The TP53 mutation profile was extracted from previously acquired whole-exome sequencing data [41]. We used the gene expression matrix normalized to transcript per million (TPM), as aforementioned, to estimate the enrichment score of every sample in these gene sets: Classical, Mesenchymal, and Proneural GBM groups, according to Verhaak [11]. These gene sets were downloaded from the Molecular Signatures Database (MSigDB) [61]. The enrichment score was assessed with the Bioconductor R package Gene Set Variation Analysis (GSVA) v1.40.1. Expression data on the FAS Exo8 Del variant (NM_152872) were extracted for PDCL 1^Sens^ and PDCL 12^Res^ under basal conditions.

### 4.5. RT-qPCR

For FAS and TNFR2 expression analysis of PDCLs in response to **P5**, PDCLs were exposed to 0.38 and 5 µM for 24 h. The Universal Probe Library (UPL) system was used for RT-qPCR with the following primers and probes (Table 2):

### 4.6. Protein Array

For apoptosis protein analysis, PDCL 1^Sens^ was exposed to 0.38 µM **P5** for 24 h. Proteins were extracted and an apoptosis protein array (Abcam, Amsterdam, The Netherlands) dot blot was performed according to the manufacturer’s instructions.

### 4.7. Statistics

All statistical tests were performed using GraphPad Prism 6. Chi-square tests and Pearson’s correlations were performed to compare the transcriptomic subgroups and IC50s. Analysis for identification of signaling pathways (*p*-values and activation z-score) from RNA sequencing differential analysis was performed using Ingenuity Pathway Analysis software (Qiagen) available from [62]. For all other analyses, one-way ANOVA with Tukey’s test for multiple comparisons or a two-tailed *t*-test for single comparisons was performed on the mean ± s.e.m.

## 5. Conclusions

Our study of the effects of six ferrocifens on the 15 GBM PDCLs led us to classify these compounds into three groups (G1–3) according to the correlation of their IC_50_ values relative to **P5**, currently the most studied ferrocifen, and to divide the PDCLs into sensitive or resistant to ferrocifens. Group G1 included **P5** plus the three other complexes bearing the ferrocenyl–double bond–phenol chain motif. This confirms the prominent role played by this motif in their cytotoxicity. In this group, the cytotoxicity of the complexes varied substantially between PDCLs, despite all these PDCLs belonging to the same cancer type, GBM.

Our in-depth studies showed that the PDCLs sensitive to **P5** were all of the Mesenchymal and Proneural subtype, while the PDCLs resistant to **P5** were mainly of the Classical subtype. This study has also allowed us to advance our understanding of the mechanism of action of **P5**. We showed that **P5** activated the Death Receptor signaling pathway in sensitive PDCLs and acted via the modulation of the expression of FAS, but its effect was not influenced by *TP53* mutations that are frequently found in GBM [8].

The identification of the behaviors of **P15** and **P41**, which are unique and different from those of the other complexes, is particularly interesting. It confirms that **P15** can have very different behavior from the compounds of the G1 group, even though all possess the ferrocenyl–double bond–phenol motif. The tamoxifen-like amino chain appeared to eliminate the differences observed among the various PDCLs and conferred the homogeneously high cytotoxicity of **P15** against all PDCLs. We also propose here that in some of the PDCLs tested, the large amount of ROS generated by **P41** was the driving force of its cytotoxicity. Overall, this work reveals a greater degree of complexity in the mechanisms of action of ferrocifens than has been reported previously.

Finally, this work highlights the value of a personalized molecular analysis of these tumors, allowing those most likely to respond to ferrocifens to be selected for this treatment. It would also need to be complemented by in vivo studies, including the development of a formulation suitable for these lipophilic complexes that would allow them to pass the blood–brain barrier (BBB).

## Figures and Tables

**Figure 1 ijms-22-10404-f001:**
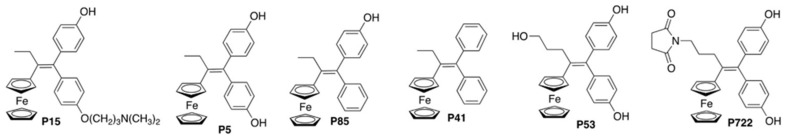
Structure of ferrocifens used in this study.

**Figure 2 ijms-22-10404-f002:**
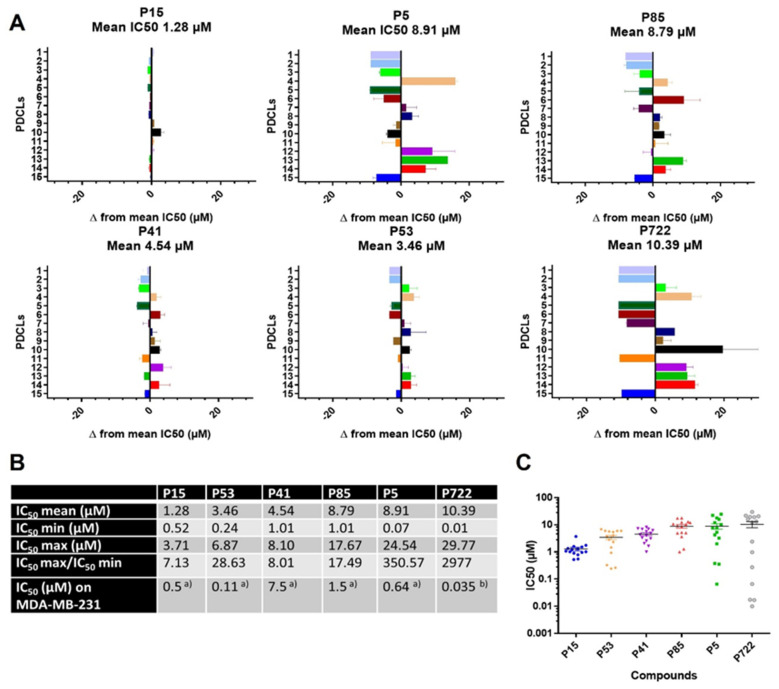
Ferrocifens induce diverse impacts on PDCL viability according to the genetic context. (**A**) For each compound, differences from the mean IC_50_ values obtained for all PDCLs. (**B**) Mean IC_50_ values obtained for each compound for all PDCLs, highest (IC_50_ max) and lowest (IC_50_ min) values and their ratio. ^a)^ IC_50_ values from [15] ^a)^ from [28] ^b)^ for MDA-MB-231 (hormone-independent breast cancer cells). (**C**) For each compound, the graphical representation of IC_50_ values was obtained for all PDCLs and their range.

**Figure 3 ijms-22-10404-f003:**
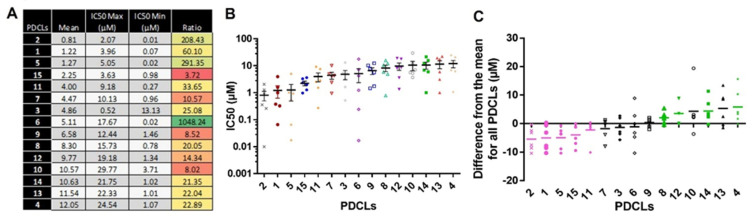
The range of response to ferrocifens differs between PDCLs. (**A**) For each PDCL, the mean IC_50_ obtained for all compounds, the highest (IC_50_ max) and lowest (IC_50_ min) and their ratio. Shades of green indicate the largest ratios, shades of red indicate the smallest ratios. (**B**) For each PDCL, the graphical representation of mean IC_50_ values for all compounds and the range is shown. (**C**) For each PDCL, the graphical representation of the difference from the mean IC_50_ value obtained for all compounds (as listed in Figure 2B) and individual values obtained for each compound are shown. PDCLs with negative values are classified as sensitive and PDCLs with positive values are classified as resistant to the ferrocifen complexes.

**Figure 4 ijms-22-10404-f004:**
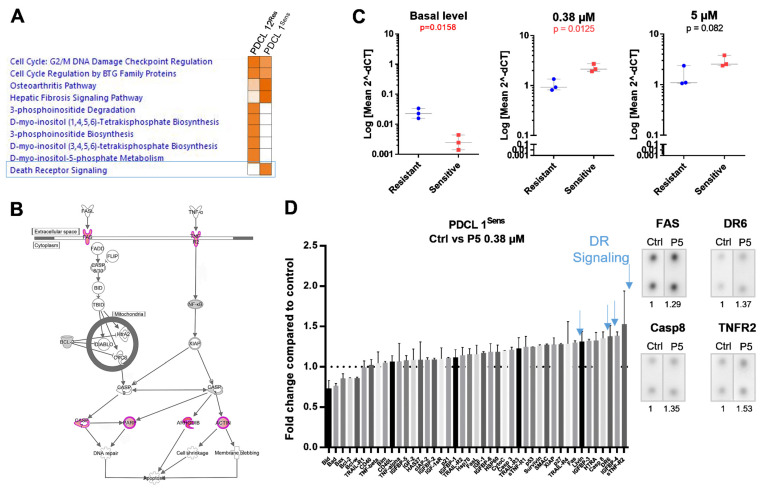
Death Receptor signaling pathway and FAS expression dynamics predict the response to compound **P5**. (**A**) Orange intensity is proportional to the level of pathway activation (IPA z-scores). The Death Receptor signaling pathway is predicted to be activated (*p*-value of 0.000127 and z-score of 2.449) in PDCL 1^Sens^ in response to **P5**, unlike in PDCL^Res^ (incubation for 24 h at their IC_30–50_ values, 0.35 and 5 µM, respectively). (**B**) Representation of the Death Receptor signaling pathway showing in red the genes that were overexpressed in PDCL 1^Sens^ in response to **P5**. (**C**) FAS expression as measured by RT-qPCR in three PDCL^Res^ (PDCLs 12, 13, and 4) and PDCL^Sens^ (PDCLs 1, 5, and 6) at the basal level or in response to 0.38 or 5 µM **P5**. (**D**) Apoptosis protein array of PDCL 1^Sens^ in response to **P5** (compared to untreated–Ctrl-cells). Representative images of dot blots of proteins involved in the Death Receptor (DR) signaling pathway for the control and **P5**-treated PDCL 1^Sens^ conditions are shown.

**Figure 5 ijms-22-10404-f005:**
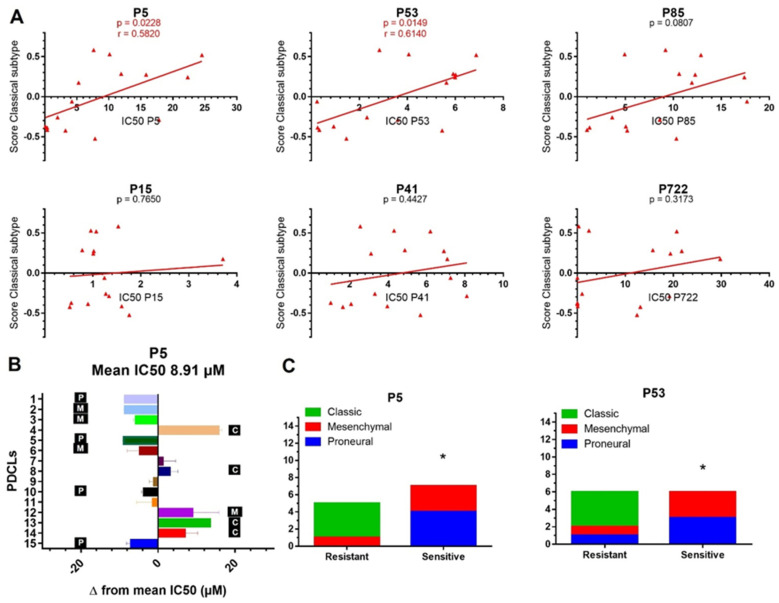
Transcriptomic subgroups predict the response to **P5** and **P53**. (**A**) Correlations between classical subgroup scores and IC_50_s for each compound. (**B**) For each PDCL, the difference from the mean IC_50_ value (8.91 µM) obtained for all PDCLs in response to P5 and their transcriptomic subgroup is shown. C = Classical, P = Proneural, M = Mesenchymal. PDCLs with bars on the right of the median line are considered resistant to P5 while PDCLs with bars on the left are considered sensitive. (**C**) Occurrence of PDCLs from each transcriptomic subgroup in the resistant group and the sensitive group in response to **P5** or **P53.** * *p*-value ≤ 0.05.

**Table 1 ijms-22-10404-t001:** List of PDCLs, their attributed ID number, TP53 gene status, and transcriptomic subtype according to the gene expression signature [11]. WT (wild type).

PDCL ID	PDCL Name	TP53 Status	Molecular Subtype	PDCL ID	PDCL Name	TP53 Status	Molecular Subtype
1	N13-1300	Pro177Ser/Lys132Asn	Proneural	9	GBM4339	Ser241Phe	Mesenchymal
2	N15-0460	Tyr220Cys	Mesenchymal	10	N15-0516	WT	Proneural
3	GBM7097	WT	Mesenchymal	11	N14-0522	WT	Classical
4	N14-1208	WT	Classical	12	N15-0385	Met237Lys	Mesenchymal
5	N14-0072	Tyr220Cys	Proneural	13	GBM6240	WT	Classical
6	N16-0535	Arg273His	Mesenchymal	14	GBM4371	Arg175His	Classical
7	N15-0661	WT	Classical	15	N13-1520	Arg248Gln	Proneural
8	N14-1525	WT	Classical				

**Table 2 ijms-22-10404-t002:** Primers and probes used for the RT-qPCR.

Gene	Forward Primer	Reverse Primer	UPL
FAS	GTGGACCCGCTCAGTACG	TCTAGCAACAGACGTAAGAACCA	#60
TNFR2	CTCCTTCCTGCTCCCAATG	CACACCCACAATCAGTCCAA	#23
PPIA	ATGCTGGACCCAACACAAAT	TCTTTCACTTTGCCAAACACC	#48

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
