# Peer review of "Heterogeneity of Response to Iron-Based Metallodrugs in Glioblastoma Is Associated with Differences in Chemical Structures and Driven by FAS Expression Dynamics and Transcriptomic Subtypes"

_ijms, 2021, doi:10.3390/ijms221910404_

Round 1

Reviewer 1 Report

This is a very interesting and well written paper that extends the authors' interest in ferrocifens to investigations of their cytotoxicity in Glioblastoma.

The selection of the specific ferrocifens used in this study is well justified and the analysis based on structure/activity relationships is sound. The data is well presented.

I have only a couple of comments/questions:

  1. The iron complexes are all dissolved in DMSO and then added to the cell matrix. Do the complexes remain in solution? Do the complexes pass into the cells to equal extents?
  2. Line 300 says that "the status of TP53, ..., is irrelevant." But on line 326 there is a discussion of the effect of some of the iron complexes on TP53 mutants PDCLs, which seems to contradict the statement on line 300. Can the authors clarify this?

Author Response

Answers to Reviewer 1:

First of all, we would like to thank you for your positive comments on our paper.

Here are our answers to your questions: 

Q1: The iron complexes are all dissolved in DMSO and then added to the cell matrix. Do the complexes remain in solution? Do the complexes pass into the cells to equal extents?

A1: We have previously quantified, by ICP-MS, the amount of iron found in cells after incubation in the presence of 30 µM of P15 and we have found that P15 significantly accumulates in cells and preferentially in the nucleus and mitochondria (Scalcon et al, Metallomics, 2017, 9, 949). Ferrocifens are highly lipophilic molecules as evidenced by their high LogPo/w values (values in the range 4.3 - 6.4 for the selected molecules). As a consequence, they rapidly accumulate in cells in in vitro experiments. With a such high lipophilicity values no difference is expected between the selected molecules. This clarification has been added in the experimental part.

Q2:  Line 300 says that "the status of TP53, ..., is irrelevant." But on line 326 there is a discussion of the effect of some of the iron complexes on TP53 mutants PDCLs, which seems to contradict the statement on line 300. Can the authors clarify this?

A2: We thank you for pointing out this apparent contradiction. Accordingly, line 300 has been changed as follows: “When looking at the molecular context present in PDCLs sensitive to ferrocifens, it appears that the presence of mutations on TP53, a key regulator of apoptosis, does not block the effect of ferrocifens, and suggests that TP53-dependent apoptosis pathways are not crucial in the response to ferrocifens.”

Comment: English language and style are fine/minor spell check required

English language has been reviewed by a native English speaker

Reviewer 2 Report

The paper by Vessières and coworkers describes the characterization of a group of ferrocifenes as antitumor agents in different patient-derived cell lines, also disclosing the biochemical pathway involved in the antitumor activity of one of the tested molecules. The paper is well written and presented and results described are in good accordance with the experimental output. Despite the scientific merit of the paper there are few issues to be discussed from my side:

1) The authors have fully investigated one of the synthesized molecules, namely P5, showing selectivity over Proneural PDLC. The first question is: how this selectivity correlates with the molecular pathway involved in P% pharmacological activity? This issue should be wider discussed.

2) Similarly, why only partial selectivity over mesenchymal PDLC is shown by P5? Could the author provide explanation of this behavior?

3) It is unclear to me why the authors decided to deepen the investigation concerning P5 (that is effective only on a single PDLC subtype) without investigating the molecular pathways underlying the strong efficacy of derivative P15, that exerts antitumor efficacy over all the PDLC subtypes, particularly mesenchymal subtype, the one associated with the worst prognosis possible. Please explain. P15 chemically differs from the other ferrocifenes, thus any potential difference in pharmacological activity should rely on the side chains. Some further investigations about would increase the overall quality of the paper.

Author Response

Answers to Reviewer 2

First of all, we would like to thank you for your positive comments on our paper.

Here are our answers to your questions: 

Q1 :  The authors have fully investigated one of the synthesized molecules, namely P5, showing selectivity over Proneural PDLC. The first question is: how this selectivity correlates with the molecular pathway involved in P5 pharmacological activity? This issue should be wider discussed.

A1: We thank you for this pertinent question. We briefly discussed the possible mechanisms of resistance in the Classical subtype on line 325 of our initial submission (Hence, in our PDCLs from the Classical subtype, it is possible that RTK signaling altered the balance of apoptosis modulating proteins towards a state of resistance to FAS-induced cell death). One can only speculate that in Proneural PDCLs, mechanisms regulating FAS-induced cell death are more functional than their Classical counterparts. A sentence was added in line 327 to suggest this: “Conversely, mechanisms regulating FAS-induced cell death in sensitive PDCLs from the Proneural and Mesenchymal subtypes may be less influenced by RTK signaling and thus more functional.

Q2) Similarly, why only partial selectivity over mesenchymal PDLC is shown by P5? Could the author provide explanation of this behavior?

A2: Thank you for raising this interesting question.  In an attempt to address it, we looked at the expression of the FAS Exo8 Del alternative FAS variant, which is known to block FAS-mediated apoptosis (see reference Cascino, G Papoff, R De Maria, R Testi, G Ruberti: Fas/Apo-1 (CD95) receptor lacking the intracytoplasmic signaling domain protects tumor cells from Fas-mediated apoptosis, Journal of Immunology, 1996, 156,13-17). Interestingly, we saw that PDCL 12 expressed this variant, while PDCL 1 does not. However, the role of this variant in the resistance to P5 here is only speculative. We added a sentence regarding this in the discussion, line 329:

“Notably, the Mesenchymal PDCL resistant to P5 (PDCL 12) expresses the FAS Exo8 Del alternative FAS variant, which is known to block FAS-mediated apoptosis (ref Cascino et al). It is thus possible that in this PDCL, this variant may have contributed to resistance to P5 in a molecular context that would have been otherwise favorable to FAS-induced cell death.”

And in the methods, line 413:

“Expression data on the FAS Exo8 Del variant (NM_152872) was extracted for PDCL 1Sens and PDCL 12Res in basal conditions.”

Q3: It is unclear to me why the authors decided to deepen the investigation concerning P5 (that is effective only on a single PDLC subtype) without investigating the molecular pathways underlying the strong efficacy of derivative P15, that exerts antitumor efficacy over all the PDLC subtypes, particularly mesenchymal subtype, the one associated with the worst prognosis possible. Please explain. P15 chemically differs from the other ferrocifens, thus any potential difference in pharmacological activity should rely on the side chains. Some further investigations about would increase the overall quality of the paper.

A3: Thank you for this relevant question.  One of the main goals of the paper was to evidence that the biological effects of ferrocifens differ in different cell lines from the same type of cancer, here glioblastoma. This is what is observed for almost all the complexes except for P15. This is why we decided to further investigate the effect of P5 which confirms our hypothesis and shows a correlation with 2 other molecules (P53 and P722). The natural resistance observed with P5 in some PDCLs was an opportunity to study further the molecular defects present in these PDCLs compared to sensitive PDCLs. Our work does not suggest that P15, which is indeed active on all PDCLs, is not pertinent in the context of therapy. This is now underlined in line 339:Interestingly, P15 was active on all PDCLs regardless of their transcriptomic subtype or TP53 mutational status, and could constitute an effective therapeutic option for all GBM. Thus, ferrocifens provide an attractive alternative to treat a subgroup of resistant GBMs.”

Comment: Moderate English changes required

English language has been reviewed by a native English speaker

Round 2

Reviewer 2 Report

The authors replied to my concerns. I consider their reply enough clear to make the paper accepatable for publication in IJMS.